# Improving gut virome comparisons using predicted phage host information

Michael Shamash,[1] Anshul Sinha,[1] Corinne F. Maurice[1,2]

**ABSTRACT** The human gut virome is predominantly made up of bacteriophages (phages), viruses that infect bacteria. Metagenomic studies have revealed that phages in the gut are highly individual specific and dynamic. These features make it challenging to perform meaningful cross-study comparisons. While several taxonomy frameworks exist to group phages and improve these comparisons, these strategies provide little insight into the potential effects phages have on their bacterial hosts. Here, we propose the use of predicted phage host families (PHFs) as a functionally relevant, qualitative unit of phage classification to improve these cross-study analyses. We first show that bioinformatic predictions of phage hosts are accurate at the host family level by measuring their concordance to Hi-C sequencing-based predictions in human and mouse fecal samples. Next, using phage host family predictions, we determined that PHFs reduce intra- and interindividual ecological distances compared to viral contigs in a previously published cohort of 10 healthy individuals, while simultaneously improving longitudinal virome stability. Lastly, by reanalyzing a previously published metagenomics data set with >1,000 samples, we determined that PHFs are prevalent across individuals and can aid in the detection of inflammatory bowel disease-specific virome signatures. Overall, our analyses support the use of predicted phage hosts in reducing between-sample distances and providing a biologically relevant framework for making between-sample virome comparisons.

**IMPORTANCE** The human gut virome consists mainly of bacteriophages (phages), which infect bacteria and show high individual specificity and variability, complicating cross-study comparisons. Furthermore, existing taxonomic frameworks offer limited insight into their interactions with bacterial hosts. In this study, we propose using predicted phage host families (PHFs) as a higher-level classification unit to enhance functional cross-study comparisons. We demonstrate that bioinformatic predictions of phage hosts align with Hi-C sequencing results at the host family level in human and mouse fecal samples. We further show that PHFs reduce ecological distances and improve virome stability over time. Additionally, reanalysis of a large metagenomics data set revealed that PHFs are widespread and can help identify disease-specific virome patterns, such as those linked to inflammatory bowel disease.

**KEYWORDS** virome, gut microbiome, bacteriophages, bioinformatics, microbial interactions

**Peer Reviewers** Prasanth Manohar, Zhejiang University-University of Edinburgh Institute, Haining, China; Michael Joseph Tisza, National Cancer Institute, Bethesda, Maryland, USA

Address correspondence to Corinne F. Maurice, corinne.maurice@mcgill.ca.

Michael Shamash and Anshul Sinha contributed equally to this article. The author names are in alphabetical order by last name.

The authors declare no conflict of interest

The human gut virome, the collection of viruses in the human gastrointestinal tract, is dominated by bacteria-infecting bacteriophages (phages). This community is highly diverse and individual-specific at the nucleotide level (1–5). This vast diversity makes it challenging to perform cross-individual or cross-cohort comparisons, as it is rare for all individuals in a cohort group to share a single viral OTU (vOTU). Most recently, a longitudinal analysis of 59 individuals further demonstrated that the individuality of the

gut virome confounded disease signal detection in the context of inflammatory bowel diseases (IBDs) in part due to intraindividual fluctuations of viruses over time (6).

Viral clusters, such as those generated by vConTACT 2 (7), have been proposed as a potential solution to this, by grouping together viruses based on their shared protein content. While this approach is useful for complete genome sequences, it may not always be reliable in the context of virome data sets. Indeed, current short-read virome studies often generate several contigs for a single viral genome, raising the risk that each contig from a given virus would be placed into a different viral cluster, confounding ecological conclusions (7).

An important limitation of previous virome analyses was the inability to confidently link uncultured phages with their hosts. Recent experimental and bioinformatic advances aim to address this issue. For instance, proximity ligation sequencing is being used to assign phages to their hosts *in situ*. With this approach, phage DNA inside of host cells at the time of sampling is covalently crosslinked to the bacterial host DNA, leading to the generation of chimeric reads during the sequencing process (8, 9). While it may not capture all interactions, this is the most validated high-throughput method for the discovery of phage-host pairings. On the computational side, tools such as iPHoP enable the high-throughput prediction of hosts using phage sequence data alone (10). Using a combination of existing tools and machine learning models, iPHoP can consistently predict hosts down to the genus level. These two approaches, proximity ligation and iPHoP, have yet to be formally compared with each other for assigning hosts to gut virome-derived sequences.

Here, we propose using predicted phage host range to allow for ecologically relevant comparisons of viromes across individuals, regardless of nucleotide-level diversity. These comparisons could provide broad insight on ecosystem function, as phages have the ability to alter bacterial abundances and metabolism (11, 12). We introduce the term Phage Host Family (PHF) as a term to describe the predicted bacterial host of a phage sequence, at the family level. This family-level cut-off was determined based on comparisons of predicted phage host range from iPHoP with experimental assignments via proximity ligation sequencing of human and mouse fecal samples, where high concordance was seen down to the family, but not genus level. Using this metric, we then re-evaluate two previously published landmark studies, one highlighting the inter-individual differences in gut virome composition and the other representing one of the largest disease-specific data sets with virome data. First, we apply PHF analysis to viromes from a cohort of 10 healthy individuals (1), sampled longitudinally for approximately 1 year, and conclude that incorporating PHFs reduced interindividual variation, while also increasing within-individual virome stability over time. Second, we analyze the phageome of a large cohort of individuals with IBDs (13), where we determine that aggregating vOTUs using PHFs allows for the detection of greater disease-specific differences in the virome in addition to reducing interindividual variability. We propose that the use of PHFs as an ecologically informed unit of phage classification is useful in allowing for cross-sample comparisons in gut virome studies.

## MATERIALS AND METHODS

### Preparing fecal samples for proximity ligation sequencing

Two infant participants were enrolled with the informed written consent of their parents for the utilization of their samples and met specific inclusion criteria: they had no diagnosed gastrointestinal disease and had not used antibiotics in the 3 months prior to sampling. Both infants (18 and 24 months of age) were on a mixed milk/solid food diet. Fresh fecal samples were collected fresh in the morning, aliquoted in an anaerobic chamber, and kept at −70°C until processing.

Six adult female germ-free C57BL/6 mice were maintained in Tecniplast IsoCages at McGill University. Mice had unlimited access to irradiated diet (Research Diets, New Brunswick, NJ) and autoclaved water. Germ-free mice were humanized by oral gavage

of 200 µL of human donor feces resuspended in sterile-reduced PBS (0.02µm-filtered). A total of 10 mouse fecal samples (from the 6 HMA mice), and 2 human fecal samples, were collected for proximity ligation and bulk metagenome sequencing. Fresh fecal samples were collected and stored at −70°C until processing. All mice had unlimited access to standard chow and water. Fecal samples were resuspended in 1 mL PBS (0.02 µm filter-sterilized). After an initial centrifugation at 1,000 × $g$ for 1 minute to pellet large debris, the bacterial cell-containing supernatant was centrifuged again at 10,000 × $g$ for 10 minutes. Pelleted bacterial cells were resuspended in 1 mL of a PBS-formaldehyde solution (1% formaldehyde) and incubated at room temperature for 20 minutes to cross-link DNA. Glycine was added in excess to quench unused formaldehyde and incubated for 15 minutes at room temperature. The fixed bacterial cells were pelleted (10,000 × $g$ for 10 minutes) and washed twice with PBS. The final resuspended bacterial pellet was transferred to a BeadBug tube with 0.1 mm silica glass beads (Benchmark Scientific, Sayreville, United States) and vortexed at maximum speed for 5 minutes. The sample was transferred to a DNA LoBind tube and sent to Phase Genomics (Seattle, United States) for library preparation and sequencing using their ProxiMeta kit and analyzed using the corresponding bioinformatic pipeline (14). In addition, a bulk metagenome was sequenced for each sample: immediately after resuspending the original fecal sample in PBS, 250 µL of sample was used for DNA extraction using the QIAGEN PowerFecal Pro DNA kit (QIAGEN, Hilden, Germany) following the manufacturer's instructions. All libraries (proximity ligation and bulk metagenome) were sequenced using the Illumina NovaSeq platform with 2 × 150 bp reads.

## Comparing host predictions between iPHoP and proximity-ligation sequencing

Viral contigs identified by the ProxiMeta pipeline were used as input for iPHoP (v. 1.3.3) (10) to computationally predict hosts, and the output was imported into R (v. 4.2.2) for analysis. The proximity-ligation linkage data were imported into R and filtered to keep only viral contigs who also had hosts predicted with iPHoP. These two data sets were then compared for concordance at the following taxonomic ranks: phylum, class, order, family, and genus. Concordance was calculated at each taxonomic rank using three distinct approaches to account for phages which have multiple assigned/predicted hosts: (i) consider the pairing concordant if the most confident iPHoP prediction matches the top Hi-C hit (most stringent); (ii) consider the pairing concordant if the most confident iPHoP prediction matches any of the Hi-C hits; and (iii) consider the pairing concordant if any of the iPHoP predictions match any of the Hi-C hits (least stringent). The percent concordance was calculated as the number of viral contigs with concordant hosts, divided by the total number of viral contigs, and multiplied by 100%.

## Re-analysis of Shkoporov et al. data set

The published phyloseq object (1) was downloaded and imported into R (v. 4.2.2) using phyloseq (v. 1.42) (15). Published virome contigs were also downloaded and filtered to keep those with length >1 kb using seqkit (v. 2.5.1), resulting in 57,721 contigs (average length 10.1 kb, maximum length 868.5 kb, N50 15.5 kb). iPHoP (v. 1.3.3) (10) was used to predict the bacterial hosts of these contigs, and the output was imported into R for analysis. The GTDB tree used by iPHoP (bac120_r202.tree) was also imported into R using phyloseq's read_tree command and combined with the downloaded phyloseq object. Viral contig relative abundance was calculated and added to the phyloseq object, replacing the existing otu_table object. Predicted host information was added to the phyloseq object as a tax_table object. In cases where a viral contig had more than 1 predicted host by iPHoP, the most confident host prediction was selected. Taxa bar plots were generated using microshades (v. 1.10) (16). Samples with less than 30% of the community consisting of contigs with unknown hosts were retained for subsequent analysis. Phyloseq's tax_glom function was used to agglomerate viral contigs that have the same predicted host at the family level. Vegan (v. 2.6–6.1) (17) was used to calculate

distances between samples using the bray, wunifrac, and unifrac metrics. Distances were evaluated by comparison type, either inter- or intraindividual sample comparisons, and the Friedman test with post-hoc Wilcoxon signed-rank test (using Bonferroni correction for multiple comparisons) was used to test for significance.

We define virome stability as the similarity between two sequential samples, and it is calculated as follows: stability = (1 − distance from the previous sample). We calculated virome stability using distances between consecutively collected samples from the same individual using Bray-Curtis distances at the contig and PHF levels and tested for significance using the Wilcoxon signed-rank test.

## Re-analysis of the HMP2 data set

In the human IBD cohort, originally analyzed by Lloyd-Price et al., bulk metagenome reads were obtained from 1,595 samples belonging to 130 individuals (27 non-IBD, 65 CD, and 38 UC) sampled longitudinally over 1 year (13). Data were downloaded from https://ibdmdb.org/results. Paired-end sequencing reads (101 bp) were generated using Illumina HiSeq 2000 or 2500. Raw reads were trimmed based on sequence quality using Trimmomatic (v 0.33) (18). Quality-controlled sequences that aligned to human and mouse genomes were removed by Bowtie2 (v. 2.2) (19). These steps were performed using the kneaddata workflow (20). Quality-controlled reads were then grouped by individual and co-assembled into 3,249,501 contigs > 1.5 kb using MEGAHIT (v. 1.2.9) (21). The contigs within each co-assembly were classified as phages by VIBRANT (v. 1.2.1) (22). In total, there were 81,422 predicted phages across all co-assemblies. These contigs were then filtered for completeness using CheckV (v. 1.0.3) (23), keeping only the 6,741 contigs that were over 50% complete. A 50% completeness cutoff was used to balance the trade-offs of overestimating viral richness due to fragmentation during assembly and maintaining viral richness. These remaining contigs were then dereplicated using blastn, keeping contigs with an average nucleotide identity of 95% over 85% alignment fraction relative to the shorter sequence (24). The resulting dereplicated contigs were considered vOTUs (24). Similar to the analyses of the Shkoporov et al. data set, iPHoP (v. 1.3.3) (10) was used to predict the bacterial host of each phage contig, keeping the most confident host prediction if there were multiple predictions. VIBRANT was used to determine whether vOTUs were considered integrated prophages (22). Quality-controlled reads were mapped to the phage vOTU library using Bowtie2 (19). vOTUs were considered present in a given sample using mapping thresholds defined by Stockdale et al. (6), where a vOTU was present if Bowtie2 mapped reads covered 50% of contigs < 5 kb, 30% of contigs ≥ 5 kb and <20 kb, or 10% of contigs ≥ 20 kb. After calculating Good's coverage and generating rarefaction curves for each sample, 502 samples were removed which had below 1,500 viral length-normalized read counts (25). PCoAs and Bray-Curtis distances on the remaining 1,093 samples were generated using MicroViz (v. 0.12.1) (26), which uses Vegan as a wrapper. DESeq2 (v. 1.44) (27) was used to calculate differentially abundant PHFs based on dysbiosis status, using the simple formula: design = ~Participant.ID+dysbiosis_binary. PHFs with an adjusted $P$ value ≤ 0.05 and with a $\log_2$ fold-change ≥1 or with a $\log_2$ fold-change ≤ −1 were considered differentially abundant. For differential abundance analyses, only individuals which had both a dysbiotic and non-dysbiotic sample were included so that a paired analysis could be conducted. Only the 18 PHFs that were more than 50% prevalent across individuals were considered. This arbitrary threshold was used to consider only the features that were widely distributed and abundant across samples. Auxiliary metabolic genes (AMGs) were predicted from vOTUs using VIBRANT. Using KEGG annotations, VIBRANT categorizes these AMGs into metabolic categories (22). In some cases, a single AMG belonged to multiple metabolic categories. MetaPhlAn 4 (v. 4.11) was used to profile the composition of bacterial families from bulk metagenomes (28). The -t rel_ab_w_stats flag was used to obtain the estimated number of reads mapping to each bacterial family. To ensure family-level taxonomy was consistent between iPHoP PHF predictions and bacterial families, NCBI-GTDB mapping was performed (29) using a "fraction mapping" threshold

of 0.6. Spearman's rank correlation coefficients were calculated to associate the relative abundances of PHFs and bacterial families. Significant associations were identified using a $P \leq 0.05$ cutoff after applying a Benjamini-Hochberg correction. Differential abundance analyses of the bacteriome were performed using DESeq2, with the same statistical cutoffs that were used with PHFs (see above). Only bacterial families that had a corresponding prevalent PHF, as defined by being present in >50% of individuals, were included. Of the 18 prevalent PHFs, only 12 had an associated MetaPhlAn-assigned bacterial family, all of which were included in the bacterial differential abundance analyses.

## RESULTS

### Computational prediction of phage hosts is concordant with proximity ligation sequencing assignments to the family level

We conducted proximity ligation (Hi-C) sequencing on 10 fecal samples from human microbiota-associated mice and 2 fecal samples from healthy human donors. After Hi-C host assignment, we identified 1,577 phage-host pairings consisting of 1,547 phages targeting 77 unique hosts at the genus level, with some phages being linked to more than one host. Using iPHoP, we then predicted hosts for the 1,547 phages with Hi-C-assigned hosts, yielding 1,587 phage-host pairings, comprising 1,243 phages targeting 108 unique hosts at the genus level. These 1,243 phages, which had hosts assigned by both Hi-C and iPHoP, were used for subsequent comparisons between approaches.

Concordance between the two approaches was calculated at each taxonomic rank from phylum to genus using three distinct approaches to account for some phages having multiple assigned/predicted hosts: (i) pairing is concordant if the most confident iPHoP prediction matches the top Hi-C hit (most stringent); (ii) pairing is concordant if the most confident iPHoP prediction matches any of the Hi-C hits; and (iii) pairing is concordant if any of the iPHoP predictions match any of the Hi-C hits (least stringent). Regardless of the comparison approach, there was consistently over 98% concordance at the phylum level, over 97% concordance at the class level, over 96% concordance at the order level, and over 92% concordance at the family level (Fig. 1). Genus-level concordance was lower, with approximately 67% concordance using comparison metrics i and ii, and 73% concordance using metric iii (Fig. 1).

Based on these findings, we decided to use iPHoP predictions at the family level for our subsequent analyses as this was the lowest taxonomic rank which still had high (>92%) concordance between the tool's predictions and Hi-C experimental assignments. These family-level host predictions will now be referred to as PHF.

### Using predicted hosts as a functional measure of virome diversity reduces interindividual variation and increases intraindividual stability

We next wanted to use PHFs to evaluate functional virome diversity within and across individuals. Using a previously published data set consisting of 140 total samples from 10 healthy individuals (1), we predicted hosts for the provided assembled phage contigs ($n$ = 57,721 contigs). In total, iPHoP yielded 197,994 species-level host predictions for 49,852 (86.3%) of the viral contigs (an average of 3.97 bacterial hosts predicted per viral contig). The remaining 7,869 (13.7%) contigs had no host predicted. The most confident iPHoP prediction for each contig was retained and used in downstream analysis. Overall, there was a large variation in the proportion of the virome made up of contigs with known hosts (Fig. 2A). To ensure that the subsequent comparisons between samples are fair (i.e., by comparing samples with similar proportions of the community represented by contigs having assigned hosts), we filtered the data set to retain only samples which had less than 30% of the community consisting of contigs with unknown hosts, resulting in a new data set composed of 63 samples from 10 individuals. Keeping all samples, including those with a high proportion of contigs with unknown hosts, would introduce bias into our analyses by over-representing incomplete or ambiguous community

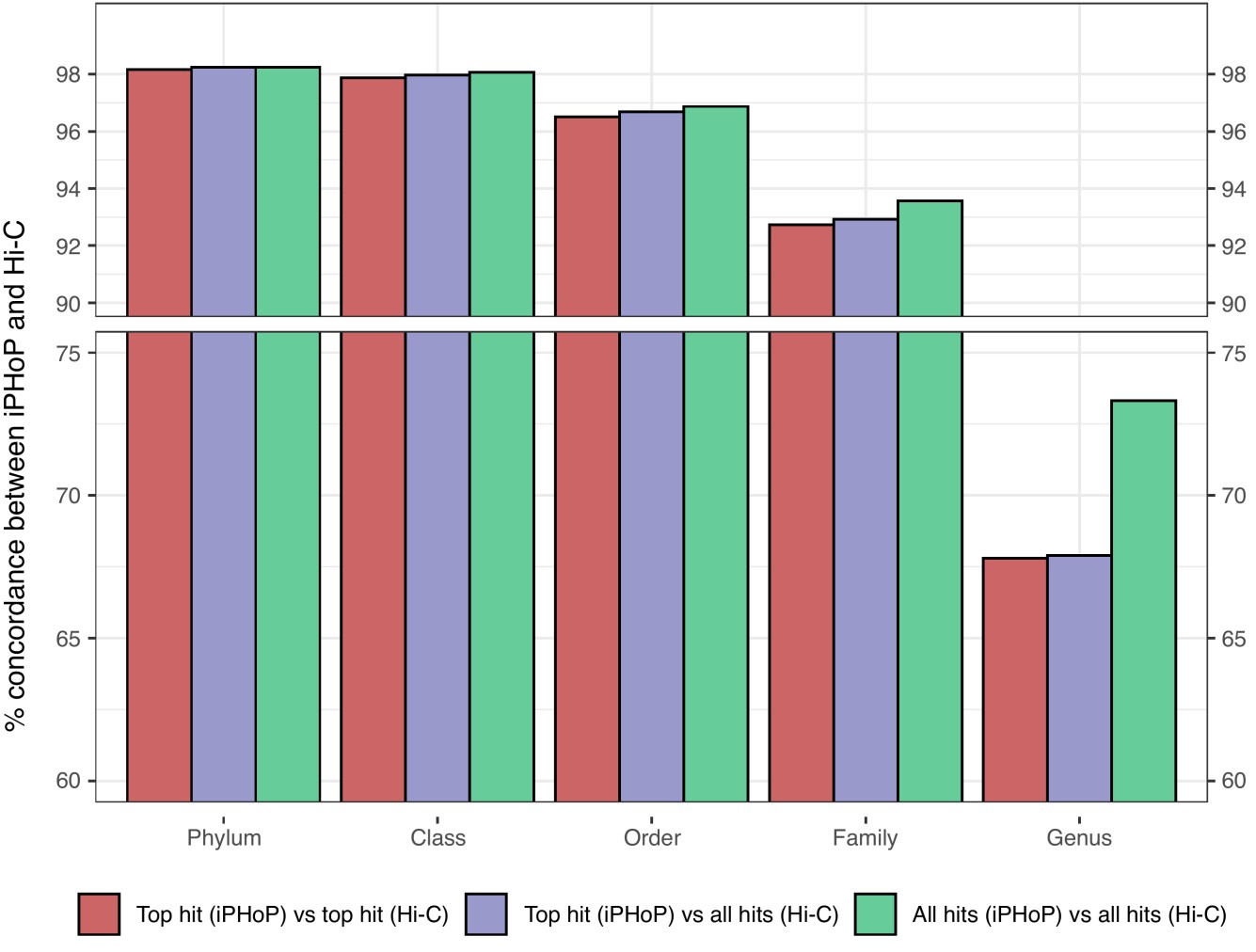

**FIG 1** Computationally predicted bacterial hosts for vOTUs are concordant with *in situ* associations to the bacterial family level. Agreement between iPHoP predicted host range and Hi-C assigned host range at various taxonomic ranks for 1,243 vOTUs. Additional comparisons were made when iPHoP predicted multiple hosts for a vOTU (see main text for details on the three comparisons).

structures, potentially leading to inaccurate conclusions about the relationships between samples. Using CheckV to filter for contigs > 50% complete did not significantly change the proportion of the community consisting of contigs with unknown hosts (Fig. S1).

Phage sequences with the same family-level host predictions were agglomerated into PHF groups, and the resulting abundance matrix was used for subsequent analyses. Pairwise distances were calculated between all samples using traditional contig-level Bray-Curtis, PHF-level Bray-Curtis, and PHF-level weighted UniFrac metrics, as described in Materials and Methods. Intraindividual sample distances were consistently lower than interindividual sample distances (Fig. 2B). Regardless of inter- or intraindividual sample comparison, PHF-level weighted UniFrac distances were significantly lower than PHF-level Bray-Curtis distances, which were themselves significantly lower than contig-level Bray-Curtis distances (Fig. 2B). Finally, we evaluated the effects of using PHF-level distances on longitudinal intraindividual virome stability, calculated as the pairwise distance between all pairs of consecutive samples from each individual. Virome stability was consistently higher when incorporating PHF-level distances using the Bray-Curtis metric (Fig. 2C).

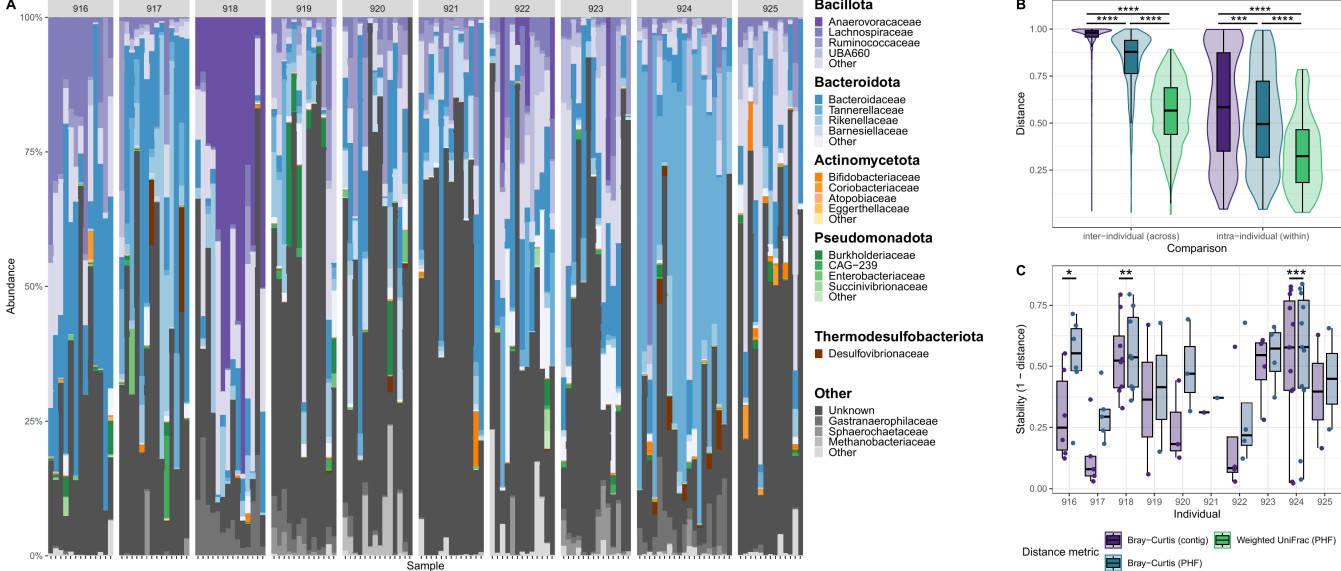

**FIG 2** PHFs reduce interindividual variation and increase intraindividual virome stability in a cohort of 10 healthy individuals. Data were analyzed from a previously published study of 10 healthy individuals (1). (A) Taxonomic bar plots of virome composition at the PHF level for each individual over time. Facet labels above the bar plots correspond to the subject IDs from the original study. (B) Ecological distances between samples with Bray-Curtis at the contig level, Bray-Curtis at the PHF level, and Weighted UniFrac at the PHF level. Interindividual and intraindividual comparisons are both shown. Significance was assessed using the Friedman test with the post-hoc Wilcoxon signed-rank test, using Bonferroni correction for multiple comparisons (***$P < 0.001$, ****$P < 0.0001$). (C) Virome stability, defined here as (1—ecological distance from the previous sample), was calculated for each individual using the Bray-Curtis distance metrics at the contig level and at the PHF level. Significance was assessed using the Wilcoxon signed-rank test (*$P < 0.05$, **$P < 0.01$, ***$P < 0.001$).

## PHFs are prevalent and can provide biological insight into the IBD virome

We next wanted to characterize PHFs in a larger data set and determine whether agglomerating at the PHF level could improve the detection of disease-specific signatures. To do so, we re-analyzed the human microbiome project 2 (HMP2) data set containing longitudinal bulk metagenome samples from IBD and non-IBD controls. After removing samples which contained low read counts (see Materials and Methods), 1,093 samples from 57 CD, 31 UC, 27 non-IBD controls remained for further analyses (68.5% of total samples) (13). From these samples, we co-assembled contigs, predicted phages using VIBRANT, filtered for phage completeness >50% using CheckV, and dereplicated contigs using 95% average nucleotide identity over 85% alignment fraction. Using this approach, we obtained a total of 3,870 distinct vOTUs across the samples within the data set. Of these 3,870 vOTUs, 23.4% (907/3,870) were VIBRANT-predicted prophages, and 87.1% (3,370/3,870) had an iPHoP predicted host family. In total, these 3,370 vOTUs belonged to 74 distinct PHFs. Interestingly, the amount of vOTUs comprising each PHF varied greatly, with some PHFs comprised of hundreds of distinct vOTUs, whereas some rare PHFs were only comprised of a single vOTU (Fig. S2).

To further characterize PHFs, and to link their host associations with metabolic functionality, we searched for viral-encoded AMGs. These genes, which are expressed throughout the process of viral infection, are thought to provide phages with increased fitness via modulation of host metabolism (22, 30). In general, there was a slight trend toward increased carriage of AMGs by prophage vOTUs (as identified by VIBRANT) compared to non-prophage vOTUs (18.5 AMGs/Mb assembled vs 12.3 AMGs/Mb assembled). In total, 45/74 PHFs carried at least 1 AMG and 12/74 PHFs carried at least 10 AMGs. In general, PHFs were enriched in amino acid metabolism, energy metabolism, and cofactor and vitamin metabolism genes (Fig. S3A), in line with previous surveys of AMGs in human microbiomes (22). Notably, compared to other PHFs, *Bacteroidaceae*-infecting phages were enriched in carbohydrate metabolism genes (Fig. S3B). *Enterobacteriaceae*-infecting phages, on the other hand, were enriched in protein folding, sorting,

and degradation genes (Fig. S3C), and in particular *cysO,* which encodes a sulfur-carrier protein important in cysteine biosynthesis and resistance to oxidative stress (31). Fourteen distinct *Enterobacteriaceae*-infecting phage vOTUs carried *cysO* (Fig. S3D). Only 3/74 other PHFs (*Pasteurellaceae*, *Pseudomonadaceae*, *Burkholderiaceae*) carried *cysO* on seven distinct contigs (Fig. S3D). Interestingly, all of these PHFs infect bacteria from the phylum Pseudomonadota, potentially reflecting host-specific adaptation through the carriage of this AMG.

Consistent with the high levels of interindividuality at the vOTU level observed in the Shkoporov et al. data set, we found that only 236/3,870 (6.10%) vOTUs were found in more than 50% of individuals in the HMP2 data set (Fig. 3A). In contrast, a higher proportion of PHFs (18/74; 24%) were found in more than 50% of individuals (Fig. 3B). Importantly, these prevalent features made up a significantly higher mean relative abundance in samples at the PHF level compared to the vOTU level (Fig. 3C). Thus, prevalent PHFs represent a larger fraction of the total community in comparison to prevalent vOTUs. In line with these observations, intraindividual and interindividual Bray-Curtis distance between samples was significantly lower at the PHF level in comparison to the vOTU level (Fig. 3D).

Given that PHFs reduced ecological distance between samples, we hypothesized that this would also allow for more biologically relevant comparisons between individuals, and ultimately a greater ability to detect disease-specific signatures in the human virome. We first generated PCoA plots using Bray-Curtis distance and found that the first two principal components explained more cumulative variance when agglomerating the virome at the PHF level in comparison to the vOTU level (Fig. 4A; 39.3% vs 11%). Importantly, the proportion of variance explained by diagnosis (non-IBD, CD, UC) was higher using PHFs than using vOTUs (Fig. S4 $R^2$= 0.0261 vs $R^2$ = 0.0185). Lloyd-Price et al. defined dysbiotic samples within this HMP2 data set as those with high microbiota divergence from non-IBD controls (13). Using this designation, we also found that dysbiosis status explained a higher proportion of variance using PHFs when compared to vOTUs (Fig. 4A; $R^2$ = 0.0394 vs $R^2$ = 0.0157). We also performed differential abundance analyses to determine whether certain PHFs were enriched or depleted depending on dysbiosis status. Including only prevalent PHFs (found in >50% of individuals), we identified a single PHF enriched in dysbiotic samples (*Enterobacteriaceae*) and four significantly depleted PHFs (*CAG-74, Ruminococcaceae, Acidaminococcaceae*, *Acutalibacteraceae*) (Table S1; Fig. 4B). These observations suggest that predicted phage hosts can be used to identify certain IBD-specific virome signatures.

To determine whether abundance and diversity trends were consistent between PHFs and their respective hosts, we assessed family-level bacterial composition using MetaPhlAn 4 (28). To ensure naming consistency, we converted taxonomic assignments between the NCBI (used by MetaPhlAn 4) taxonomic framework and GTDB (used by iPHoP) (see Materials and Methods). After doing so, we identified 33 bacterial families with an associated PHF. For each of these 33 PHF-host pairs, we determined the association of relative abundances across the data set. In a majority of the pairs (28/33), there was a significant positive association between the relative abundance of PHFs and their host families (Table S2). This is in line with previous studies that have shown strong correlations in the diversity of gut viromes and bacteriomes from the same individuals (32). Prevalent PHF/host pairs such as *Bacteroidaceae* and *Lachnospiracaeae* had particularly strong correlations ($\rho > 0.8$) (Table S2). Still, there was a range of significant positive associations ($\rho = 0.07–0.93$). For instance, prevalent PHF/host pairs *Oscillospiraceae* ($\rho = 0.6$) and *Tannerellaceae* ($\rho = 0.59$) showed more modest associations (Table S2). To further compare differences in diversity between the virome and bacteriome, we performed differential abundance analyses, this time comparing differentially abundant bacterial families (with a corresponding prevalent PHF; see Materials and Methods) between dysbiotic and non-dysbiotic samples. Three of the significantly depleted PHFs (*CAG-74, Ruminococcaceae, Acutalibacteraceae*) did not have an associated MetaPhlAn-assigned bacterial family. Similar to its corresponding PHF, *Enterobacteriaceae* was significantly

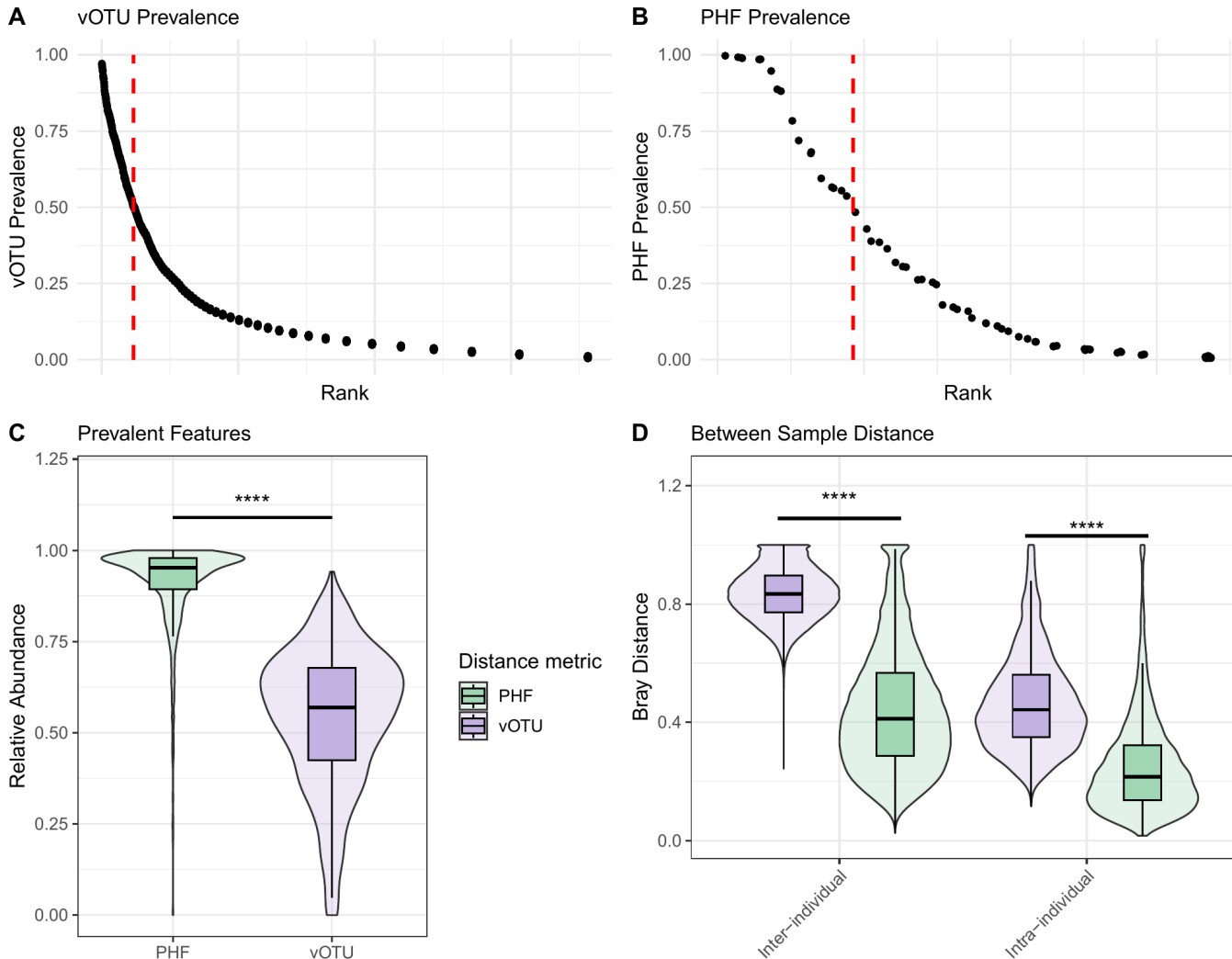

**FIG 3** PHFs are prevalent and reduce intra- and interindividuality in a large human IBD cohort. Data were analyzed from the previously published HMP2 data set (13). Samples with low viral read counts (<1,500) were removed from analyses. In total, bulk metagenomes from 1,093 samples from 115 individuals (57 CD, 31 UC, 27 non-IBD controls) were included for downstream analyses. (A, B) Rank prevalence distributions of vOTUs (A) and PHFs (B) across individuals. In total, there were 3,870 distinct vOTUs and 74 distinct PHFs. The dotted red line indicates the rank at which features are more, or less than, 50% prevalent. (C) Mean relative abundance of features (PHFs vs vOTUs) that were present in more than 50% of individuals in the data set. (D) Bray-Curtis distance between samples according to interindividual or intraindividual comparisons. Significance was assessed using the Wilcoxon signed-rank test (****$P \leq 0.0001$).

enriched in dysbiotic samples (Fig. S5; Table S1). While certain PHF/host pairs exhibited similar trends (Table S1), such as *Bacteroidaceae* and *Rikenellaceae*, this pattern was not consistent for all cases. For instance, *Oscillospiraceae* was significantly depleted, but its PHF was not (Fig. S5; Table S1). Similarly, while not significant, *Burkholderiaceae* increased to a greater extent in dysbiotic samples compared to its corresponding PHF (Table S1). Overall, while bacterial and viral family abundances are highly correlated, these associations are variable and can differ between disease states.

## DISCUSSION

In the past decade, the development of phage-specific bioinformatic tools, alongside large cohort viral metagenomic studies, has revealed key characteristics of the human gut virome. Notably, gut viromes exhibit high levels of interindividuality (1, 6) and temporal variation (6). While we can now appreciate the sheer genomic phage diversity that our collective guts harbor, it remains a challenge to understand how similar our viromes are over time and from one another. Here, through re-analysis of two landmark

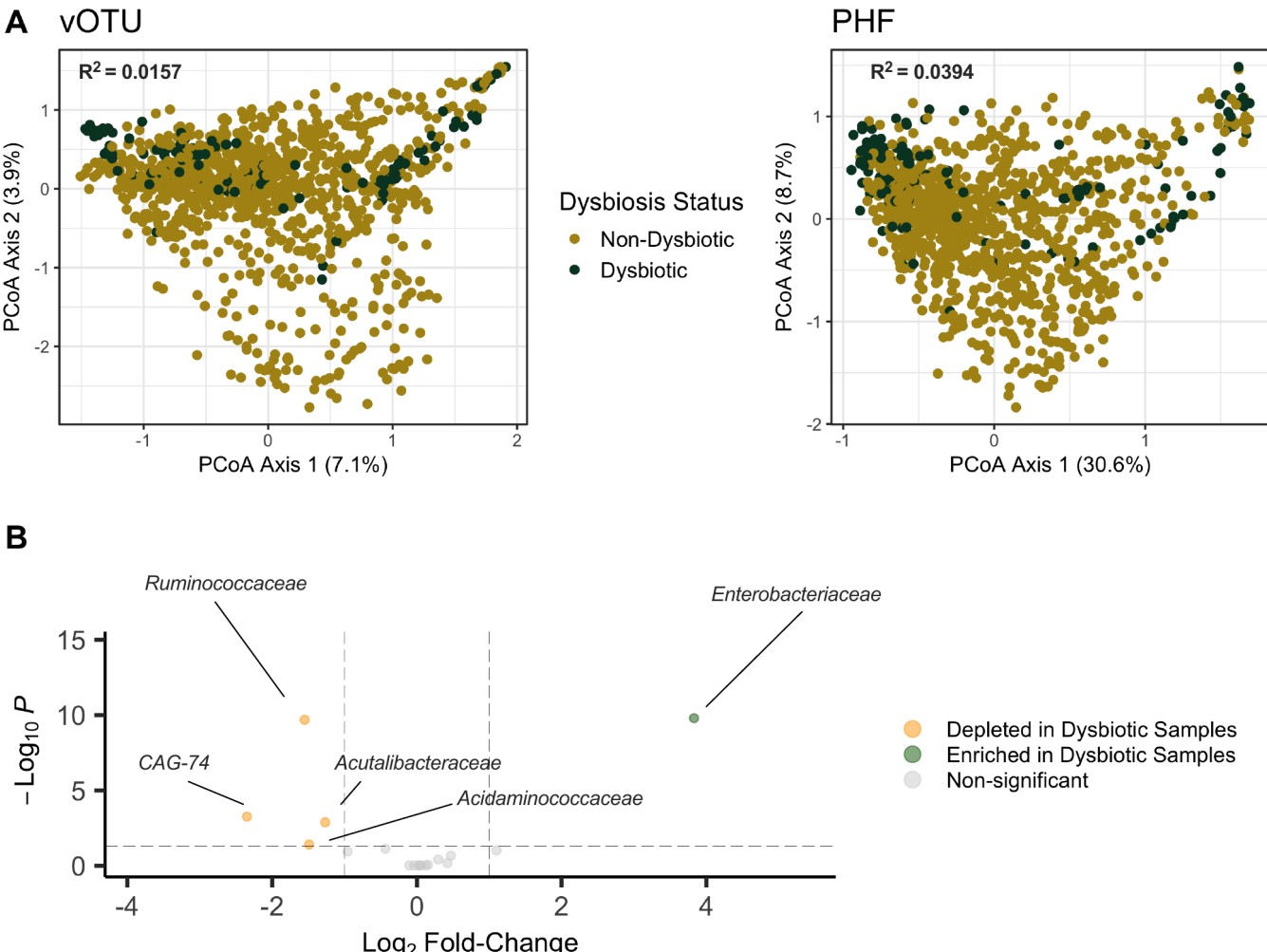

**FIG 4** PHFs reveal disease-specific signatures of IBD. Data were analyzed from the previously published HMP2 data set (13). Samples with low viral read counts (< 1,500) were removed from analyses. (A) PCoA plots generated from Bray-Curtis distance matrices using vOTUs (left) and PHFs (right). Samples are color-coded according to the dysbiotic status identified in (13). (B) Differentially abundant PHFs based on dysbiosis status. Only individuals which had both a dysbiotic and non-dysbiotic sample were included. Only PHFs that were more than 50% prevalent across individuals were considered for these analyses. PHFs with an adjusted $P$ value ≤ 0.05 and with a $\log_2$ fold-change ≥1 or with a $\log_2$ fold-change ≤ −1 were considered differentially abundant.

studies of gut viromes, we demonstrate that the use of predicted phage host families (PHFs) can improve virome comparisons between and within individuals, resulting in valuable functional information typically lost with current approaches.

Using PHFs as a unit of classification in two independent published data sets, we showed that in comparison to vOTUs, intra- and inter-personal ecological distance is reduced, indicating that despite phages differing between samples at the contig/vOTU level, their functionality remains similar. These findings are reminiscent of the functional redundancy characteristic of gut bacterial communities, whereby phylogenetic differences between individuals exist despite conserved functional profiles (33, 34). The conserved functionality of both phage and bacterial communities over time likely contributes to the stability and resilience of both subsets.

The advantages of working with reduced between-sample virome distance were evident as we showed that the first two principal coordinate axes of PCoA plots explained more variance when using PHFs as the unit of classification. We also showed that the proportion of variance explained by disease and dysbiosis status was greater when using PHFs. These findings are in line with those from Clooney et al. (2), who analyzed human IBD viromes. They showed that gene-sharing-based genus-level

taxonomy, compared to contig-based analyses, better identified disease-associated compositional changes and increased the variance explained by the first two principal coordinate axes. These observations together highlight the importance of using PHFs as a qualitative unit of phage classification when making cross-individual comparisons of gut viromes.

A key additional benefit of using predicted hosts lies in the biologically relevant information they provide. This contrasts with existing gene-sharing and phage morphology-based taxonomy approaches, where taxonomic groups are not necessarily informative of how phages interact with their bacterial hosts or the ecosystem at-large (35). Phages in several ecosystems, including the gut, have been shown to be strong regulators of bacterial abundance, diversity, and metabolism (11, 12, 36). Therefore, grouping phages by their predicted hosts provides context for the effects that they may have on the bacterial community and beyond. We showed that in the context of IBD, dysbiotic samples were enriched in *Enterobacteriaceae* PHFs and depleted in *CAG-74, Ruminococcaceae, Acidaminococcaceae,* and *Acutalibacteraceae* PHFs. Our analyses provide a framework to identify interactions relevant to disease although follow-up studies are needed to understand the importance of these phage-host interactions. For instance, phage enrichment in tandem with host depletion could be relevant to several diseases (37–39). In exploring associations between PHF and bacterial family abundance pairs, we generally observed strong positive correlations, in line with previous studies (32, 40). The strong associations between host and viral abundance are in agreement with the piggyback-the-winner hypothesis, whereby the lysogenic replication cycle is favored in environments, such as the gut, where there are high bacterial abundances (41). In cases where there are weaker associations, this could represent scenarios where there is a switch from lysogenic to lytic replication, leading to discordant abundance relationships.

The increase in *Enterobacteriaceae* PHFs we observed in dysbiotic samples is likely a consequence of increased host abundance. Still, it is interesting to note that these phages were enriched in *cys*O, a gene involved in cysteine biosynthesis. Notably, *cys*O has been directly tied to defense against oxidative stress (42). As *Enterobacteriaceae* are known to proliferate in the inflamed gut in the face of oxidative stress (43, 44), our data imply that phage-encoded AMGs could be a source of this resistance. More broadly, the observation that different PHFs carry distinct AMGs suggests that grouping at the phage host family level provides an additional layer of functional insight beyond phage-host relationships. However, an important consideration is that potential inaccuracies in defining prophage borders (45) could lead to an overestimation of AMGs (46). Thus, while these findings merit further investigation, they should be interpreted with caution.

While we propose the use of PHFs for between-sample virome comparisons, it should be noted that this method is reliant on the sensitivity of iPHoP (or any other phage-host matching bioinformatic tool used). In our analyses, between 12.9% (Lloyd-Price et al. data set) and 13.7% (Shkoporov et al. data set) of vOTUs did not have an iPHoP-assigned host. This may become an even larger issue if this approach is applied to non-human associated microbiomes where iPHoP performs with less sensitivity (10). Regardless, it is reasonable to assume that the sensitivity of bioinformatic phage-host prediction tools will improve alongside recent improvements in phage genome reconstruction approaches such as contig extension (47) and viral binning (48).

To assess the accuracy of bioinformatic phage-host predictions, we measured the concordance at different taxonomic ranks between iPHoP, a bioinformatic tool, and Hi-C sequencing, which relies on physical linkage between phage and host. Due to the prohibitive costs associated with Hi-C sequencing, especially when applied to large sample volumes, we suggest using iPHoP for family-level host predictions as a suitable alternative. Still, this approach should be interpreted with caution as the concordance between iPHoP and Hi-C sequencing was only assessed using fecal samples. These trends could feasibly differ depending on the environment sampled.

Lastly, as phage host range is often not beyond the species and strain level, by grouping phages at the host family level, this method lacks the sensitivity to detect trends in specific phage-host pairs. Despite these limitations, as computational methods to detect phage-host pairs improve their resolution, similar approaches to PHFs could be used at lower taxonomic ranks.

## AUTHOR AFFILIATIONS

[1]Department of Microbiology & Immunology, McGill University, Montreal, Quebec, Canada
[2]McGill Centre for Microbiome Research, Montreal, Quebec, Canada

## AUTHOR ORCIDs

Michael Shamash http://orcid.org/0000-0001-7900-647X
Anshul Sinha http://orcid.org/0000-0003-4745-3851
Corinne F. Maurice http://orcid.org/0000-0001-7187-3472

## AUTHOR CONTRIBUTIONS

Michael Shamash, Data curation, Formal analysis, Investigation, Writing – original draft, Writing – review and editing, Conceptualization | Anshul Sinha, Conceptualization, Data curation, Formal analysis, Investigation, Writing – original draft, Writing – review and editing | Corinne F. Maurice, Funding acquisition, Supervision, Writing – review and editing

## DATA AVAILABILITY

Code used for data analysis is available at https://github.com/mshamash/PHF_manuscript. Whole genome and Hi-C sequencing reads are available in the NCBI SRA under accession number PRJNA1145458.

## ETHICS APPROVAL

Human fecal samples were collected with the approval of protocol A04-M27-15B from the McGill University Institutional Review Board. Mouse fecal samples were collected from human microbiota-associated (HMA) mice with the approval of McGill University animal use protocol MCGL-7999.

## ADDITIONAL FILES

The following material is available online.

### Supplemental Material

**Supplemental Figures (mSystems01364-24-S0001.pdf).** Figures S1 to S5.
**Legends (mSystems01364-24-S0002.docx).** Legends for the supplemental figures and tables.
**Supplemental Table 1 (mSystems01364-24-S0003.csv).** Differentially abundant PHFs and bacterial families.
**Supplemental Table 2 (mSystems01364-24-S0004.csv).** PHF and host family abundance correlations.

### Open Peer Review

**PEER REVIEW HISTORY (review-history.pdf).** An accounting of the reviewer comments and feedback.

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
