## [Reviewer comments · mSystems]

Improving gut virome comparisons using predicted phage host information

Michael Shamash, Anshul Sinha, and Corinne Maurice

Corresponding Author(s): Corinne Maurice, McGill University Faculty of Medicine and Health Sciences

Review Timeline:

Submission Date:	October 11, 2024
Editorial Decision:	December 20, 2024
Revision Received:	February 13, 2025
Accepted:	February 28, 2025

Editor: Evelien Adriaenssens

Reviewer(s): Disclosure of reviewer identity is with reference to reviewer comments included in decision letter(s). The following individuals involved in review of your submission have agreed to reveal their identity: Prasanth Manohar (Reviewer #1); Michael Joseph Tisza (Reviewer #3)

Transaction Report:

DOI: <https://doi.org/10.1128/msystems.01364-24>

Re: mSystems01364-24 (Improving gut virome comparisons using predicted phage host information)

Dear Dr. Corinne F Maurice:

Editor comments:

The authors use the term taxonomy when referring to their system of phage-host families. To avoid confusion with existing and official virus taxonomy, I ask the authors to change this to classification throughout the manuscript (as also recommended by the reviewers). The ICTV has recognised that there can be many different and useful ways of classification, but that there is only one official virus taxonomy (see Simmonds et al, 2023, PLoS biology).

Further, I suggest to include the additional analyses on the bacterial community as recommended by reviewer 3. While this would normally be a major revision, I hope it can be done relatively easily.

Revision Guidelines

Sincerely,
Evelien Adriaenssens
Editor
mSystems

Reviewer #1 (Comments for the Author):

The research article by Shamash et al., titled "Improving gut virome comparisons using predicted phage host information", describes a new approach to match phage virome sequences to their hosts. The authors approached the hypothesis using viral contigs from new subjects and from previous studies. The study also addressed the important question of detecting disease-specific virome markers. The reliability of PHFs on iPHoP is more limited.

Line no. 76: Need more details about the participants? Especially dieting, and sampling time.

Line no. 85: Fecal sampling is confusing. What is meant by ten samples from 6 mice? Were they all humanized, as claimed above?

Line no. 82: Resuspended in what?

Line no. 125: What was the size of the contigs?

What is the rationale for choosing Shkoporov et al. and Lloyd-Price et al.

Line no. 208: How high?

Line no. 218: Is it genus level?

Line no. 231: It should be a phage sequence, not phage.

Line no. 299: There is a more non-significant dataset to raise questions on this conclusion.

Reviewer #2 (Comments for the Author):

The manuscript by Shamash and Sinha et al. introduces a framework for evaluating the impact of viruses on the microbiome by examining phage-bacterial host relationships. Due to the high genomic diversity among phages, direct comparisons within and between viromes are often challenging. To address this, the authors propose using phage host families (PHF)-groupings based on predicted phage-host interactions-as functional units for virome analysis. Using two publicly available datasets, the authors demonstrate that PHFs reduce complexity by lowering inter- and intra-individual variation and improving phageome stability within individuals. Although only a subset of available data could be used for phageome comparisons, PHFs retained a larger proportion of the dataset for functional analysis than traditional approaches. Consequently, the authors could show that PHF increased the variance explained by disease type and microbiome diversity compared to classical vOTUs. The potential to gain additional insights into how phages influence and modulate the microbiome, particularly when including additional accessory genes besides AMG, is exciting.

Minor comments:

Could the authors clarify how many vOTUs identified in the HMP2 project were classified as proviruses by VIBRANT?

Additionally, is there a trend in AMG encoding between virulent and temperate phages (Line 263)?

Line 20: Referring to PHF as a "unit of phage taxonomy" may be misleading, given the current focus on basing phage taxonomy on genomic identity. It is very likely that the viruses in the PHF share little genome identity with each other. How about calling it a 'unite of phage classification' instead?

In the abstract (line 16), the authors state that the diversity of phages makes meaningful cross-study comparisons difficult. Can the PHF approach effectively address this issue?

In the PHF-Hi-C comparison (Line 192), some phages are noted to have multiple hosts, suggesting a broader host range. How does PHF handle phages with a broader host range in the benchmarking dataset? Do they all fall into the same PHF group since phages are grouped at the host family level, or are they dropped and not included in a potential downstream analysis?

Fig 2C: Consider replacing the violin plots with box plots and overlaying individual data points. This would improve clarity, especially for individuals with few samples, where density plots might misrepresent data due to small sample sizes.

Supplementary Figure 1: Could you include all the family names in the PHF? Additionally, color coding the dots according to a range of vOTUS count could increase the plot's information (e.g., singleton, <10 vOTUs per PHF, 10-50 vOTUs, etc.)

Line 160: To increase clarity, states that the described dereplication resulted in the formation of vOTUs.

Line 163: Please clarify in the method section whether vOTUs in the Shkoporov et al. dataset were also agglomerated into PHFs based on host family-level classifications, as done for contigs.

Line 228-230: Why not include the graph in the supplementary data when the analysis has already been performed, and the result is used as a supporting argument in the manuscript?

Reviewer #3 (Comments for the Author):

Shamash et al identify a variety of related challenges in conducting gut virome (phageome, practically) studies and propose a

solution for improving analysis by devising a category called Phage Host Families to group virus sequences by (predicted) bacterial host family.

I applaud the authors for trying to make more sense out of metagenomic phage signals than is possible with the current paradigms. The methods used are mostly technically sound and are well described.

Major critiques:

Overall, I worry that this approach is merely recapitulating the signal that one would get from bacteria grouped by family level. After all, it's plausible that when grouping at the family level, the abundance of bacteria and their phages would be extremely highly correlated.

To address this, the authors could run Metaphlan4 on the HMP2 IBD data and use the family level relative abundances. This repo should help convert taxonomy between the NCBI (metaphlan) scheme and GTDB (used by iphop): https://github.com/nick-youngblut/gtdb_to_taxdump. Use script `ncbi-gtdb_map.py`

Then, do the bacterial and PHF approaches agree? If not, what might account for differences?

Next, if this approach is to be adopted by others in the community, there should be some sort of tool or pipeline that bioinformatics scientists can use to go from predicted phages to PHFs for their data. The linked GitHub is sufficient for reproducing the specific analyses of this manuscript, but I don't see a pipeline to make PHFs.

Finally, in the abstract and perhaps elsewhere, PHFs are referred to as a "higher level taxonomic category". This seems wrong to me. PHF is a separate type of qualitative category. Please justify this statement or change it for accuracy.

Minor critiques:

In lines 292-295, the statement about "we also found that dysbiosis status explained a higher proportion of variance" should be rephrased or better supported. It refers to Fig. 4A, but I don't see why this panel explains or measures this claim. Is there a panel missing?

While concordance of iphop and Hi-C drops at the genus level, it's not clear that Hi-C should be considered ground truth data.

AMG identification (Supp Fig 2) should be considered tenuous at best considering the contigs are from bulk metagenomes. Often, phages integrate next to AMGs and the prophage border is not properly determined by available computational methods. What is the rate of AMG detection in bulk metagenomes (HMP2) compared to the virome assembly dataset used earlier (Shkoporov)?

Improving gut virome comparison using predicted phage host information

Authors: Michael Shamash, Anshul Sinha, and Corinne F. Maurice

The manuscript by Shamash and Sinha et al. introduces a framework for evaluating the impact of viruses on the microbiome by examining phage-bacterial host relationships. Due to the high genomic diversity among phages, direct comparisons within and between viromes are often challenging. To address this, the authors propose using phage host families (PHF)—groupings based on predicted phage-host interactions—as functional units for virome analysis. Using two publicly available datasets, the authors demonstrate that PHFs reduce complexity by lowering inter- and intra-individual variation and improving phageome stability within individuals. Although only a subset of available data could be used for phagosome comparisons, PHFs retained a larger proportion of the dataset for functional analysis than traditional approaches. Consequently, the authors could show that PHF increased the variance explained by disease type and microbiome diversity compared to classical vOTUs. The potential to gain additional insights into how phages influence and modulate the microbiome, particularly when including additional accessory genes besides AMG, is exciting.

Minor comments:

Could the authors clarify how many vOTUs identified in the HMP2 project were classified as proviruses by VIBRANT? Additionally, is there a trend in AMG encoding between virulent and temperate phages (Line 263)?

Line 20: Referring to PHF as a "unit of phage taxonomy" may be misleading, given the current focus on basing phage taxonomy on genomic identity. It is very likely that the viruses in the PHF share little genome identity with each other. How about calling it a 'unite of phage classification' instead?

In the abstract (line 16), the authors state that the diversity of phages makes meaningful cross-study comparisons difficult. Can the PHF approach effectively address this issue?

In the PHF—Hi-C comparison (Line 192), some phages are noted to have multiple hosts, suggesting a broader host range. How does PHF handle phages with a broader host range in the benchmarking dataset? Do they all fall into the same PHF group since phages are grouped at the host family level, or are they dropped and not included in a potential downstream analysis?

Fig 2C: Consider replacing the violin plots with box plots and overlaying individual data points. This would improve clarity, especially for individuals with few samples, where density plots might misrepresent data due to small sample sizes.

Supplementary Figure 1: Could you include all the family names in the PHF? Additionally, color coding the dots according to a range of vOTUS count could increase the plot's information (e.g., singleton, <10 vOTS per PHF, 10-50 vOTUs, etc.)

Line 160: To increase clarity, states that the described dereplication resulted in the formation of vOTUs.

Line 163: Please clarify in the method section whether vOTUs in the Shkoporov et al. dataset were also agglomerated into PHFs based on host family-level classifications, as done for contigs.

Line 228-230: Why not include the graph in the supplementary data when the analysis has already been performed, and the result is used as a supporting argument in the manuscript?

Editor comments (Evelien Adriaenssens)

The authors use the term taxonomy when referring to their system of phage-host families. To avoid confusion with existing and official virus taxonomy, I ask the authors to change this to classification throughout the manuscript (as also recommended by the reviewers). The ICTV has recognised that there can be many different and useful ways of classification, but that there is only one official virus taxonomy (see Simmonds et al, 2023, PLoS biology).

Further, I suggest to include the additional analyses on the bacterial community as recommended by reviewer 3. While this would normally be a major revision, I hope it can be done relatively easily.

We thank the editor for their consideration of our manuscript. We have revised our description of PHFs to use the term ‘classification’ instead of ‘taxonomy’. We have addressed the points raised by reviewers and have attached our point-by-point response to comments below.

Reviewer 1

The research article by Shamash et al., titled "Improving gut virome comparisons using predicted phage host information", describes a new approach to match phage virome sequences to their hosts. The authors approached the hypothesis using viral contigs from new subjects and from previous studies. The study also addressed the important question of detecting disease-specific virome markers. The reliability of PHFs on iPHoP is more limited.

We thank the reviewer for their comments and have made the necessary changes and clarifications to the main text.

Line no. 76: Need more details about the participants? Especially dieting, and sampling time.

This is now provided on L80-83.

Line no. 85: Fecal sampling is confusing. What is meant by ten samples from 6 mice? Were they all humanized, as claimed above?

We apologize for the confusion. Yes, all mice were humanized (human microbiota-associated/HMA mice) and this has been clarified in the methods (lines 84-90). Ten fecal samples were collected from these 6 HMA mice – some mice were sampled twice to reach this number (each sample collected on a different day).

Line no. 82: Resuspended in what?

This was added.

Line no. 125: What was the size of the contigs?

We now include this information on L131-132.

What is the rationale for choosing Shkoporov et al. and Lloyd-Price et al.

The work by Shkoporov *et al.*, was foundational in the field for understanding inter-individual differences in gut virome composition. We therefore deemed it important to investigate inter-individuality through the lens of PHFs using this same dataset.

The work by Lloyd-Price *et al.*, represents one of the largest disease-specific datasets that contains virome data. Since viromes have been studied extensively in the context of

inflammatory bowel disease (1–3), we also saw it fit to use an IBD-specific dataset. While the reviewer is correct in assessing that the disease-specific changes to the virome are not drastic, there still are notable differences between “dysbiotic” and “non-dysbiotic” samples. Additionally, rather than seeking a dataset that we thought would give us the most significant differences between control and disease, we analyzed what we saw as the most ideal dataset for the PHF comparisons.

We now mention the landmark aspect of these studies in our Introduction (L66-67) and in the Discussion (L351).

Line no. 208: How high?

In this case, over 92% concordance (L252).

Line no. 218: Is it genus level?

No, this is species-level and we now specify this on L261.

Line no. 231: It should be a phage sequence, not phage.

This was added.

Line no. 299: There is a more non-significant dataset to raise questions on this conclusion.

We have edited the corresponding sentence.

Reviewer 2

The manuscript by Shamash and Sinha et al. introduces a framework for evaluating the impact of viruses on the microbiome by examining phage-bacterial host relationships. Due to the high genomic diversity among phages, direct comparisons within and between viromes are often challenging. To address this, the authors propose using phage host families (PHF)-groupings based on predicted phage-host interactions-as functional units for virome analysis. Using two publicly available datasets, the authors demonstrate that PHFs reduce complexity by lowering inter- and intra-individual variation and improving phageome stability within individuals. Although only a subset of available data could be used for phagosome comparisons, PHFs retained a larger proportion of the dataset for functional analysis than traditional approaches. Consequently, the authors could show that PHF increased the variance explained by disease type and microbiome diversity compared to classical vOTUs. The potential to gain additional insights into how phages influence and modulate the microbiome, particularly when including additional accessory genes besides AMG, is exciting.

We thank the reviewer for their overall positive feedback.

Minor comments:

Could the authors clarify how many vOTUs identified in the HMP2 project were classified as proviruses by VIBRANT? Additionally, is there a trend in AMG encoding between virulent and temperate phages (Line 263)?

907 prophages of the 3,870 vOTUs were identified by VIBRANT as predicted prophages. This is now clarified in the main text (L 275). For our AMG analyses, we included the 3,370 vOTUs that had a family-level host prediction. Of these, 1,089 vOTUs (32.3%) contained at least one AMG. Of these 1,089 vOTUs, 374 (34.3%) were VIBRANT-predicted prophages. There may be a slight trend towards increased rate of AMG carriage in prophage vs. non-prophage vOTUs, with a slight increase in AMGs/Mb assembled (18.5 vs. 12.3). This is now mentioned on line 286.

An additional note on this comment: When calculating then AMGs/Mb for prophage vs. non-prophage vOTUs, we noticed a minor error in our previous calculations for AMGs/Mb of carbohydrate metabolism genes and protein folding and sorting genes. Originally, when we calculated AMGs/Mb, we summed the contig lengths of all contigs containing AMGs. However, we realized this sum should include all contigs, regardless of whether they contain AMGs or not. We thus recalculated our AMGs/Mb ratios which are now more reflective of “AMG-enrichment” for each PHF. The observation that *Enterobacteriaceae* is enriched in protein-folding genes still holds when calculated this way. However, we now see that *Bacteriodaceae* is the most enriched in carbohydrate metabolism genes of all PHFs rather than *Bifidobacteriaceae*. These new results are included in Figure S3B and Figure S3C, as well as in the main text.

Line 20: Referring to PHF as a "unit of phage taxonomy" may be misleading, given the current focus on basing phage taxonomy on genomic identity. It is very likely that the viruses in the PHF share little genome identity with each other. How about calling it a 'unit of phage classification' instead?

As suggested by the Editor, we have revised our description of PHFs to include the term "classification" instead of "taxonomy".

In the abstract (line 16), the authors state that the diversity of phages makes meaningful cross-study comparisons difficult. Can the PHF approach effectively address this issue?

As highlighted in our study (Figure 2B, Figure 3D) and others (4, 5), the gut virome is highly individual-specific at the viral contig/vOTU level, more so than the gut bacteriome. Furthermore, geographic location has also been shown to have a significant effect on virome composition (6).

Conducting cross-study comparisons further compounds issues of individuality and geography, resulting in only a fraction of the viral sequences being detected across studies. This type of data structure may make it difficult to draw conclusions, as one needs to tease apart interindividual & geographic effects from the variables under study (disease status, treatment, etc.).

Although we don't explicitly use the PHF approach here to conduct cross-study comparisons, we would argue that use of this approach can improve such comparisons. The current state of the art for virome analyses involves generating a database of vOTUs, and mapping reads from each sample back to this database. By agglomerating vOTUs instead into functional groupings based on predicted bacterial host information, more viral sequences could be used for cross-study comparisons, considering the much lower variation and heterogeneity in gut bacteriomes across individuals (5–7).

In the PHF-Hi-C comparison (Line 192), some phages are noted to have multiple hosts, suggesting a broader host range. How does PHF handle phages with a broader host range in the benchmarking dataset? Do they all fall into the same PHF group since phages are grouped at the host family level, or are they dropped and not included in a potential downstream analysis?

We thank the reviewer for this point. As we describe in the methods (L132-133, 164-166), the PHF approach only considers the most confident iPHoP prediction in cases where a single phage sequence has multiple hosts. This confidence value is returned in the final iPHoP output files. In this case, any other less-confident host predictions for a given phage sequence are dropped and not considered in downstream analysis.

Our decision to do this is twofold:

- (1) There is a longstanding assumption in the literature that 1 phage infects 1 host. While this dogma is now beginning to be challenged, researchers often still often report 1 predicted host per phage in metagenomic studies.
- (2) This is required for the downstream ecological analyses, where phage sequences are grouped by PHF (if a phage had 2 or more PHFs, then we could double- or triple-count this phage's abundance in some diversity metrics or analyses).

Ultimately, this can be seen as a limitation of the PHF approach in its current state, however as phage-host prediction tools improve, researchers may wish to consider other approaches for dealing with multiple host matches (*e.g.*, using voting approaches or considering all host matches equally in their final analysis).

Nevertheless, we explored the benchmarking dataset further to evaluate the frequency of multi-family predictions for the same phage sequence.

Only 252 phage contigs (out of 1,587 total phage contigs, so 16%) had >1 genus-level predicted host. Of these 252 phage contigs, 231 (91.7%) of them had the same family-level host prediction, 20 (7.9%) of them had 2 different family-level host predictions, and 1 (0.4%) of them had 3 different family-level host predictions.

Of the 21 contigs which had >1 family-level host prediction, 15 (71%) of them were from within the same order.

Thus, these data support our choice of using family-level host predictions for our PHFs. Despite sometimes having multiple genus-level host predictions, these phage contigs were predicted to have the same host at the family level in most cases.

Fig 2C: Consider replacing the violin plots with box plots and overlaying individual data points. This would improve clarity, especially for individuals with few samples, where density plots might misrepresent data due to small sample sizes.

We have changed the figure accordingly.

Supplementary Figure 1: Could you include all the family names in the PHF? Additionally, color coding the dots according to a range of vOTUS count could increase the plot's information (*e.g.*, singleton, <10 vOTS per PHF, 10-50 vOTUs, etc.)

We've updated the figure (now Figure S2 in the revised version) accordingly to include labels for all PHFs and dots colour-coded according to singletons, 2-10, 11-50, 51-100, and > 100 vOTU counts.

Line 160: To increase clarity, states that the described dereplication resulted in the formation of vOTUs.

This change has now been made on L165. Where relevant, “vOTUs” now replace “contigs” for the rest of the Methods section of the re-analyses of the HMP2 dataset.

Line 163: Please clarify in the method section whether vOTUs in the Shkoporov et al. dataset were also agglomerated into PHFs based on host family-level classifications, as done for contigs.

The Shkoporov *et al.* dataset and analysis of this dataset was done at the viral contig level, using the contigs and read mapping statistics provided in the original paper. These viral contigs were not clustered into vOTUs, unlike the contigs in the HMP2 dataset which were clustered into vOTUs prior to downstream analysis. As mentioned in the main text (L135-137), contigs in the Shkoporov *et al.* dataset were indeed agglomerated into PHFs based on host family-level predictions from iPHoP.

Line 228-230: Why not include the graph in the supplementary data when the analysis has already been performed, and the result is used as a supporting argument in the manuscript?

We thank the reviewer for this suggestion and have added this to the revised manuscript (Figure S1).

Reviewer 3

Shamash et al identify a variety of related challenges in conducting gut virome (phageome, practically) studies and propose a solution for improving analysis by devising a category called Phage Host Families to group virus sequences by (predicted) bacterial host family.

I applaud the authors for trying to make more sense out of metagenomic phage signals than is possible with the current paradigms. The methods used are mostly technically sound and are well described.

We thank the reviewer for their supportive comments.

Major critiques:

Overall, I worry that this approach is merely recapitulating the signal that one would get from bacteria grouped by family level. After all, it's plausible that when grouping at the family level, the abundance of bacteria and their phages would be extremely highly correlated.

To address this, the authors could run Metaphlan4 on the HMP2 IBD data and use the family level relative abundances. This repo should help convert taxonomy between the NCBI (metaphlan) scheme and GTDB (used by iphop): https://github.com/nick-youngblut/gtdb_to_taxdump. Use script `ncbi-gtdb_map.py`

Then, do the bacterial and PHF approaches agree? If not, what might account for differences?

We thank the reviewer for their thoughtful suggestion. We followed the suggestion of running MetaPhlAn and converting taxonomy. We should note that after running MetaPhlAn 4, of the 312 bacterial families detected, only 72 had an associated GTDB taxonomy. Despite this, we were able to compare most of the prevalent bacterial families and PHFs.

We correlated the relative abundances of PHFs and their associated host families by obtaining Spearman's correlation coefficients for each PHF/host pair. We also performed differential abundance analyses on bacterial host families and compared how differentially abundant families compare to their corresponding PHF. These analyses are now found at the end of the results section and in Table S2 and Figure S5. These data are also contextualized in the discussion section (L397-403).

In general, in line with previous studies (8, 9), we observed strong positive correlations between most PHFs and host family abundances. Still, there were cases where these positive correlations in prevalent PHF/host pairs were more modest ($\rho \sim 0.6$). We have included representative families that represent high, modest, and poor positive correlations in the plots below (**Response**

Figure 1). The complete Spearman’s correlation coefficients can also be found in the table below (**Response Table 1**).

Additionally, in some PHF/host pairs, we observed similar changes between dysbiotic and non-dysbiotic samples in both members of the pair, whereas in other cases these trends were discordant.

Overall, while it appears that PHFs and host family abundances are strongly correlated, the same signals are not fully recapitulated. The high concordance between bacterial and phage diversity is thought to be due to high rates of lysogenic replication in the gut (10). Thus, in families and disease states where we do not observe concordance between abundances, this could be due to a change in replication strategies. During lytic replication for instance, we’d expect higher PHF abundance compared to host. In contrast, as increased host density is associated with higher rates of lysogeny (10), it is possible that an increase in lysogens would prevent lytic replication through superinfection immunity (10), resulting in low phage:host ratios. While purely speculative, the above scenarios represent plausible explanations as to why PHF and host abundances would not necessarily agree.

It is difficult to determine why certain PHFs/host family pairs would correlate better than others. It has been recently documented that “carrier-state” replication and phase variation of surface receptors fosters phage-host coexistence in *Bacteroides* (11, 12), which could explain the strong PHF/host abundance correlations for *Bacteroidaceae* we observe. For other pairs, future studies will be needed to determine whether there are specific replication strategies or phage-defense determinants that could explain these dynamics.

Response Figure 1. Variable positive correlations between PHF and host family abundance. Representative strong (*Bacteroidaceae*), modest (*Tannerellaceae*) and weak (*Barnesiellaceae*) correlations. Correlations were quantified using Spearman's rank correlation coefficient (ρ).

Response Table 1. List of correlations between PHF and host family relative abundances. Correlations and *p* values were quantified using Spearman's rank correlation coefficient (ρ). *P* values were adjusted using the Benjamini-Hochberg method.

Family	Spearman's correlation coefficient	Adjusted p value
Akkermansiaceae	0.925397629	0
Dysgonomonadaceae	0.886670016	0
Acidaminococcaceae	0.875618171	0
Bacteroidaceae	0.861122054	6.52166652510445e-322
Lachnospiraceae	0.810535997	6.58E-255
Bifidobacteriaceae	0.792698007	5.24E-236
Marinifilaceae	0.757872282	6.51E-204
Enterobacteriaceae	0.743766421	1.86E-192
Rikenellaceae	0.714038577	1.47E-170
Burkholderiaceae	0.709442495	1.84E-167
Desulfovibrionaceae	0.680374506	5.86E-149
Oscillospiraceae	0.601829633	3.15E-108
Tannerellaceae	0.588199604	2.65E-102
Pasteurellaceae	0.415931109	1.36E-46
Pseudomonadaceae	0.313894246	4.48E-26
Coriobacteriaceae	0.268950251	2.98E-19
Selenomonadaceae	0.26287843	1.91E-18
Barnesiellaceae	0.235153062	6.19E-15
Fusobacteriaceae	0.230287705	2.21E-14
Enterococcaceae	0.226704727	5.47E-14
Peptoniphilaceae	0.160242144	1.58E-07
Lactobacillaceae	0.15814209	2.22E-07
Streptococcaceae	0.153520167	4.88E-07
Eggerthellaceae	0.119654768	0.000100674
Actinomycetaceae	0.104705834	0.000694187
Staphylococcaceae	0.096136951	0.001855909
Christensenellaceae	0.09406888	0.002261083
Muribaculaceae	0.073329556	0.018051645
Clostridiaceae	0.058691972	0.059626466
Cyanobiaceae	-0.003556444	0.906508351
Moraxellaceae	-0.004629503	0.905946132
Flavobacteriaceae	-0.011334496	0.753863967
Rhodobacteraceae	-0.023797312	0.475077881

Next, if this approach is to be adopted by others in the community, there should be some sort of tool or pipeline that bioinformatics scientists can use to go from predicted phages to PHFs for their data. The linked GitHub is sufficient for reproducing the specific analyses of this manuscript, but I don't see a pipeline to make PHFs.

We thank the reviewer for their suggestion. Although we propose a new method for virome analysis, this analysis can be easily done by connecting existing tools (iPHoP, R, phyloseq, vegan, etc.) and most likely would not require the development of a new tool. Nevertheless, we added additional example code to the GitHub repository to allow other researchers to quickly process their iPHoP output data and get an overview of PHF membership.

Finally, in the abstract and perhaps elsewhere, PHFs are referred to as a "higher level taxonomic category". This seems wrong to me. PHF is a separate type of qualitative category. Please justify this statement or change it for accuracy.

As suggested by the Editor, we have revised our description of PHFs to include the term "classification" instead of "taxonomy".

Minor critiques:

In lines 292-295, the statement about "we also found that dysbiosis status explained a higher proportion of variance" should be rephrased or better supported. It refers to Fig. 4A, but I don't see why this panel explains or measures this claim. Is there a panel missing?

We apologize for the confusion. We were originally referring to the R-squared value from the adonis PERMANOVA test, which is the percentage of variance in the distance matrix explained by a variable (in this case, dysbiosis and disease status). Thus, in Figure 4A, we were comparing the R-squared value of "dysbiosis status" using PHFs and using vOTUs. The increase in variance explained using PHFs suggests a greater difference in virome composition based on dysbiosis status than if we were using vOTUs to make a distance matrix and resulting PCoA.

To clarify our statement, we now include the R-squared values directly on the PCoA plots in Figure 4A. We also include the R-square values in Figure S4 (previously Figure S3).

While concordance of iphop and Hi-C drops at the genus level, it's not clear that Hi-C should be considered ground truth data.

We agree with the reviewer on this point, yet would argue that Hi-C is currently the closest approximation to ground truth because it captures phage-host interactions *in situ*. It remains the most validated high-throughput experimental method for *de novo* host assignment from complex

samples (13–15). While culturing thousands of phage-hosts would be ideal for generating ground-truth data, it is not currently feasible considering the time and resources required, as well as technical limitations including the limited culturability of certain sample types, especially those from the human gut (16).

AMG identification (Supp Fig 2) should be considered tenuous at best considering the contigs are from bulk metagenomes. Often, phages integrate next to AMGs and the prophage border is not properly determined by available computational methods. What is the rate of AMG detection in bulk metagenomes (HMP2) compared to the virome assembly dataset used earlier (Shkoporov)?

We recognize that detecting AMGs from bulk metagenome datasets comes with certain limitations, including challenges in accurately identifying prophage boundaries and distinguishing viral contigs from bacterial genomic sequences. To address these issues in our analyses of the HMP2 data, we used CheckV to assess viral completeness and retained only contigs classified as >50% complete.

When presenting the HMP2 bulk metagenome dataset in the manuscript, we report that 45/74 PHFs (61%) carry at least 1 AMG, and 12/74 PHFs (16%) carry at least 10 AMGs. We proceeded with the same analysis on the Shkoporov *et al.* virome assembly dataset, finding that 72/169 PHFs (43%) carry at least 1 AMG, while 22/169 PHFs (13%) carry at least 10 AMGs.

The slightly higher proportion of AMG-containing PHFs in the HMP2 dataset may stem from differences in datasets and sample type, as bulk metagenomes include a broader range of microbial and viral sequences compared to virome-specific assemblies. Importantly, the AMGs detected in both datasets include genes with predicted functions relevant to host-virus interactions, supporting the biological significance of these results.

As the number of virome and bulk metagenome studies continue to grow, incorporating long-read sequencing, alongside the refinement of existing (pro)phage detection tools, could further help to address these challenges.

References

1. Clooney AG, Sutton TDS, Shkoporov AN, Holohan RK, Daly KM, O'Regan O, Ryan FJ, Draper LA, Plevy SE, Ross RP, Hill C. 2019. Whole-virome analysis sheds light on viral dark matter in inflammatory bowel disease. *Cell Host Microbe* 26:764–778.
2. Federici S, Kredo-Russo S, Valdés-Mas R, Kviatcovsky D, Weinstock E, Matiuhin Y, Silberberg Y, Atarashi K, Furuichi M, Oka A, Liu B, Fibelman M, Weiner IN, Khabra E, Cullin N, Ben-Yishai N, Inbar D, Ben-David H, Nicenboim J, Kowalsman N, Lieb W, Kario E, Cohen T, Geffen YF, Zelcbuch L, Cohen A, Rappo U, Gahali-Sass I, Golembo M, Lev V, Dori-Bachash M, Shapiro H, Moresi C, Cuevas-Sierra A, Mohapatra G, Kern L, Zheng D, Nobs SP, Suez J, Stettner N, Harmelin A, Zak N, Puttagunta S, Bassan M, Honda K, Sokol H, Bang C, Franke A, Schramm C, Maharshak N, Sartor RB, Sorek R, Elinav E. 2022. Targeted suppression of human IBD-associated gut microbiota commensals by phage consortia for treatment of intestinal inflammation. *Cell* 185:2879-2898.e24.
3. Sinha A, Li Y, Mirzaei MK, Shamash M, Samadfam R, King IL, Maurice CF. 2022. Transplantation of bacteriophages from ulcerative colitis patients shifts the gut bacteriome and exacerbates the severity of DSS colitis. *Microbiome* 10:105.
4. Reyes A, Haynes M, Hanson N, Angly FE, Heath AC, Rohwer F, Gordon JI. 2010. Viruses in the faecal microbiota of monozygotic twins and their mothers. *Nature* 466:334–338.
5. Nishijima S, Nagata N, Kiguchi Y, Kojima Y, Miyoshi-Akiyama T, Kimura M, Ohsugi M, Ueki K, Oka S, Mizokami M, Itoi T, Kawai T, Uemura N, Hattori M. 2022. Extensive gut virome variation and its associations with host and environmental factors in a population-level cohort. *Nat Commun* 13:5252.

6. Zuo T, Sun Y, Wan Y, Chan FKL, Miao Y, Ng SC. 2020. Human-Gut-DNA Virome Variations across Geography, Ethnicity, and Urbanization. *Cell Host Microbe* 1–11.
7. Shkoporov AN, Clooney AG, Sutton TDS, Ryan FJ, Daly KM, Nolan JA, McDonnell SA, Khokhlova EV, Draper LA, Forde A, Guerin E, Velayudhan V, Ross RP, Hill C. 2019. The human gut virome is highly diverse, stable, and individual specific. *Cell Host Microbe* 26:527–541.
8. Moreno-Gallego JL, Chou SP, Di Rienzi SC, Goodrich JK, Spector TD, Bell JT, Youngblut ND, Hewson I, Reyes A, Ley RE. 2019. Virome diversity correlates with intestinal microbiome diversity in adult monozygotic twins. *Cell Host Microbe* 25:261–272.
9. Shah SA, Deng L, Thorsen J, Pedersen AG, Dion MB, Castro-Mejía JL, Silins R, Romme FO, Sausset R, Jessen LE, Ndela EO, Hjelmsø M, Rasmussen MA, Redgwell TA, Leal Rodríguez C, Vestergaard G, Zhang Y, Chawes B, Bønnelykke K, Sørensen SJ, Bisgaard H, Enault F, Stokholm J, Moineau S, Petit M-A, Nielsen DS. 2023. Expanding known viral diversity in the healthy infant gut. *Nat Microbiol* <https://doi.org/10.1038/s41564-023-01345-7>.
10. Silveira CB, Luque A, Rohwer F. 2021. The landscape of lysogeny across microbial community density, diversity and energetics. *Environ Microbiol* 23:4098–4111.
11. Porter NT, Hryckowian AJ, Merrill BD, Fuentes JJ, Gardner JO, Glowacki RWP, Singh S, Crawford RD, Snitkin ES, Sonnenburg JL, Martens EC. 2020. Phase-variable capsular polysaccharides and lipoproteins modify bacteriophage susceptibility in *Bacteroides thetaiotaomicron*. *Nat Microbiol* <https://doi.org/10.1038/s41564-020-0746-5>.

12. Shkoporov AN, Khokhlova EV, Stephens N, Hueston C, Seymour S, Hryckowian AJ, Scholz D, Ross RP, Hill C. 2021. Long-term persistence of crAss-like phage crAss001 is associated with phase variation in *Bacteroides intestinalis*. *BMC Biol* 19:163.
13. Yaffe E, Relman DA. 2020. Tracking microbial evolution in the human gut using Hi-C reveals extensive horizontal gene transfer, persistence and adaptation. *Nat Microbiol* 5:343–353.
14. Marbouty M, Thierry A, Millot GA, Koszul R. 2021. MetaHiC phage-bacteria infection network reveals active cycling phages of the healthy human gut. *eLife* 10:e60608.
15. Uritskiy G, Press M, Sun C, Huerta GD, Zayed AA, Wiser A, Grove J, Auch B, Eacker SM, Sullivan S, Bickhart DM, Smith TPL, Sullivan MB, Liachko I. 2021. Accurate viral genome reconstruction and host assignment with proximity-ligation sequencing. *bioRxiv* <https://doi.org/10.1101/2021.06.14.448389>.
16. Lagier JC, Dubourg G, Million M, Cadoret F, Bilen M, Fenollar F, Levasseur A, Rolain JM, Fournier PE, Raoult D. 2018. Culturing the human microbiota and culturomics. *Nat Rev Microbiol* 16:540–550.

Re: mSystems01364-24R1 (Improving gut virome comparisons using predicted phage host information)

Dear Dr. Corinne F Maurice:

Your manuscript has been accepted, and I am forwarding it to the ASM production staff for publication. Your paper will first be checked to make sure all elements meet the technical requirements. ASM staff will contact you if anything needs to be revised before copyediting and production can begin. Otherwise, you will be notified when your proofs are ready to be viewed.

Sincerely,

Evelien Adriaenssens
Editor
mSystems